# A computational workflow for the expansion of heterologous biosynthetic pathways to natural product derivatives

Jasmin Hafner [1,4], James Payne[2,4], Homa MohammadiPeyhani [1], Vassily Hatzimanikatis [1✉] & Christina Smolke [2,3✉]

Plant natural products (PNPs) and their derivatives are important but underexplored sources of pharmaceutical molecules. To access this untapped potential, the reconstitution of heterologous PNP biosynthesis pathways in engineered microbes provides a valuable starting point to explore and produce novel PNP derivatives. Here, we introduce a computational workflow to systematically screen the biochemical vicinity of a biosynthetic pathway for pharmaceutical compounds that could be produced by derivatizing pathway intermediates. We apply our workflow to the biosynthetic pathway of noscapine, a benzylisoquinoline alkaloid (BIA) with a long history of medicinal use. Our workflow identifies pathways and enzyme candidates for the production of (S)-tetrahydropalmatine, a known analgesic and anxiolytic, and three additional derivatives. We then construct pathways for these compounds in yeast, resulting in platforms for de novo biosynthesis of BIA derivatives and demonstrating the value of cheminformatic tools to predict reactions, pathways, and enzymes in synthetic biology and metabolic engineering.

[1] Laboratory of Computational Systems Biotechnology, Swiss Federal Institute of Technology (EPFL), Lausanne, Switzerland. [2] Department of Bioengineering, Stanford University, Stanford, CA, USA. [3] Chan Zuckerberg Biohub, San Francisco, CA, USA. [4]These authors contributed equally: Jasmin Hafner, James Payne. ✉email: vassily.hatzimanikatis@epfl.ch; csmolke@stanford.edu

Plants synthesize a remarkable range of complex and valuable molecules, known as plant natural products (PNPs), commonly used as flavors, fragrances, and medicines[1]. However, production of these molecules via extraction from plant biomass is often limited by slow growth, low yield, laborious extraction and purification procedures, and variability due to weather and climate change. Furthermore, while many modern medicines are natural products, a higher fraction are derivatives of natural products[2]. The range of PNP derivatives accessible to researchers is typically limited to those that can be readily produced via chemical synthesis from PNPs extracted from plants, while many more derivatives could potentially be made via regioselective enzymatic modification of PNPs and their intermediates. Microbial production of PNPs can potentially address these concerns, and additionally facilitates production of novel PNP derivatives by leveraging the genetic tractability of well-established microbial hosts to alter the heterologous biosynthetic pathway.

Since the landmark production of the antimalarial drug precursor artemisinic acid in *Saccharomyces cerevisiae* in 2006[3], there has been an increase in the size and complexity of pathways reconstructed in heterologous hosts[4]. This progress is highlighted by the recent de novo biosynthesis in *S. cerevisiae* of noscapine[5], an antitussive benzylisoquinoline alkaloid (BIA) and potential chemotherapeutic[6–8] from *Papaver somniferum* separated by 16 enzymatic steps from tyrosine. In that study, halogenated derivatives of tyrosine were fed to the engineered yeast strains to produce halogenated derivatives of noscapine intermediates. However, the non-native halogenated substrates were not tolerated as well as the native substrates of the pathway enzymes, and derivatives of only early intermediates in the pathway were detected. In such cases, an alternative strategy would be required to produce derivatives of more chemically complex downstream pathway intermediates or of noscapine itself.

An alternative approach to produce derivatives of PNPs and their intermediates is to integrate additional enzymes into microorganisms expressing heterologous PNP biosynthetic pathways. Enzymes that are able to accept and functionalize intermediates or products along a PNP pathway would thus produce novel products in vivo from the natural precursors. However, producing new-to-nature compounds necessarily entails the use of enzymes outside their natural functions, and in many cases an enzyme will not be known a priori with the desired non-native function. Given the wealth of enzymatic knowledge that has been accumulated, a computational method to predict enzymes that may catalyze a desired transformation will greatly expedite the development of biosynthetic pathways engineered to produce new-to-nature products.

Computational methods have been employed to guide the discovery of enzymatic functions and the design of biosynthetic pathways for the production of molecules with interesting pharmaceutical or industrial properties[9]. These methods generate hypothetical pathways to compounds of interest by assuming that enzymes that perform similar, but not identical, reactions to those desired might be promiscuous or sufficiently evolvable to perform the desired reaction after engineering and/or optimization. The concept of substrate promiscuity is translated into generalized enzymatic reaction rules that mathematically describe the reactive site recognized by an enzyme as well as the molecular rearrangement performed during the biotransformation. Popular cheminformatic tools[9–11] for predictive biochemistry include BNICE.ch (Biochemical Network Integrated Computational Explorer)[12], enviPath[13], GEM-Path[14], novoPathFinder[15], NovoStoic[16], ReactPRED[17], RetroPath2.0[18], and Transform-MinER[19]. These tools have typically been used in retrobiosynthesis studies, where the aim is to determine potential bioproduction pathways by

biochemically walking back from a target compound to the native metabolism of a chassis organism[20–22] via predicted enzymatic reaction steps. The prediction of novel reactions is subsequently followed by the search for suitable enzymes that can catalyze the predicted step. Enzyme prediction tools such as BridgIT[23], EC-BLAST[24], E-zyme[25], and Selenzyme[26] determine the structural similarity of a novel reaction to all well-characterized reactions in biochemical databases, and propose a list of enzyme candidates ranked by their likelihood to catalyze the desired transformation.

Here, we develop a computational workflow to identify potential derivatives of intermediates of a given biosynthetic pathway and subsequently predict enzyme candidates that may carry out the desired transformation(s) (Fig. 1). Our workflow expands the chemical space around a pathway of interest using BNICE.ch to create a map of all compounds accessible with known or predicted biochemical reactions and then identifies enzymes capable of carrying out the desired transformations on the prioritized set of compounds using the enzyme prediction tool BridgIT. We apply this workflow to the reconstructed noscapine biosynthetic pathway in yeast. We narrow our search to enzyme candidates capable of producing (*S*)-tetrahydropalmatine, a PNP found in plants of the genus *Corydalis* that has been shown to possess analgesic and anxiolytic effects and has shown promise as a potential treatment for opiate addiction[27–29]. After experimental evaluation of seven of the top enzyme candidates in yeast strains engineered to produce the noscapine biosynthetic intermediate (*S*)-tetrahydrocolumbamine de novo, we identify two enzymes that enabled production of (*S*)-tetrahydropalmatine. We then apply our workflow to identify three additional derivatives of pathway intermediates, predict enzymes for their biosyntheses, and then construct *S. cerevisiae* strains to produce these three products de novo. As the number of reconstructed heterologous pathways for PNPs continues to increase, we anticipate that the described workflow can be used to produce many chemically complex compounds spanning diverse therapeutic activities.

## Results

**Computational expansion of the noscapine pathway reveals thousands of potential target molecules.** Each biosynthetic pathway presents an opportunity to produce numerous derivative compounds by chemically modifying functional groups of the pathway product and its intermediates. Computational reaction prediction tools, such as BNICE.ch, allow rapid exploration of the hypothetical chemical space of potential pathway derivatives. Their generalized enzymatic reaction rules mimic known enzymatic activities in silico by recognizing and transforming a specific functional group on a substrate to generate a product. Iterative application of these rules to biosynthetic pathway intermediates creates a reaction network to hypothetical derivatives of all pathway intermediates, offering targets for bioproduction.

We applied this computational expansion process on the noscapine pathway, which starts from (*S*)-norcoclaurine and involves 17 metabolites connected by 17 reactions (Fig. 2). BNICE.ch expanded the network around the 17 metabolites for four generations, generating both known and novel reactions to produce compounds known to any biological[30–39], bioactive[40,41], or chemical[42] database. This expansion yielded a network spanning 4838 compounds (Supplementary Data 1) and 17,597 reactions (Supplementary Data 2). As our analysis focused on BIAs, we required the substrate and product to contain the minimal elemental composition of the 1-benzylisoquinoline scaffold (i.e., at least 16 carbon atoms, 13 hydrogen atoms, and 1 nitrogen atom). The resultant trimmed BIA network spanned 1518 compounds, of which 99 were classified as biological or

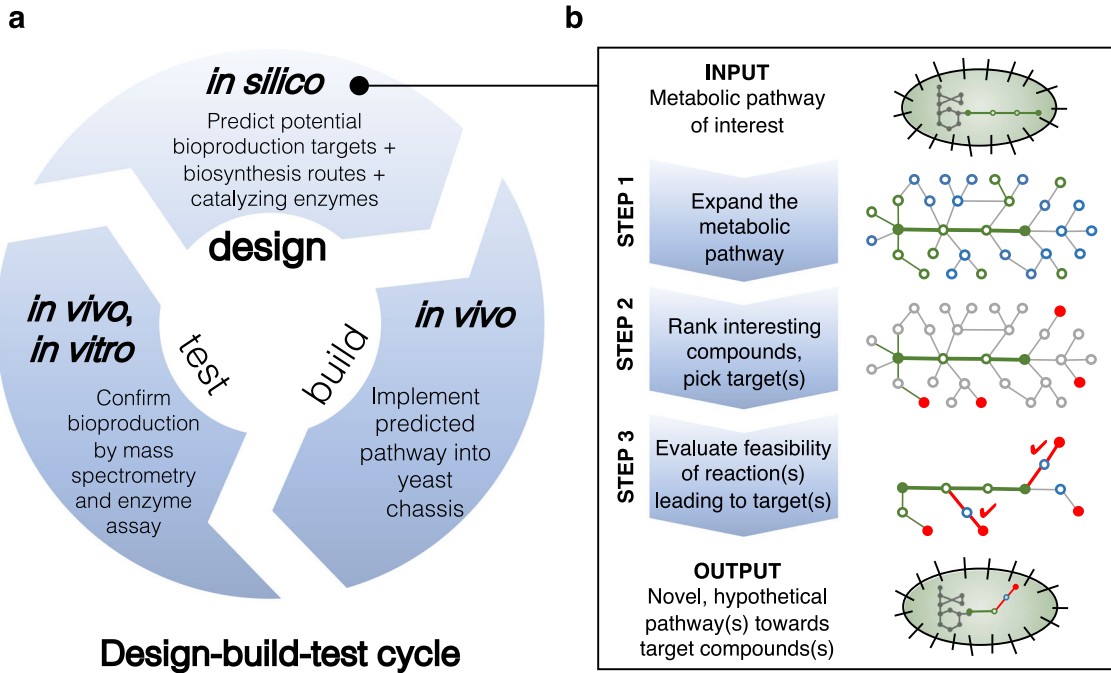

**Fig. 1 Overall workflow integrating computational prediction of target compounds, pathways, and enzymes with experimental validation. a** Applied design-build-test cycle. **b** Computational workflow. Circles represent compounds, edges represent biotransformations. Green is used to designate known biological reactions and compounds and blue circles are compounds from the chemical space without specific biological annotation. Red circles show compounds selected for their popularity in scientific literature and in the patent landscape, and red edges represent their corresponding biosynthesis pathways.

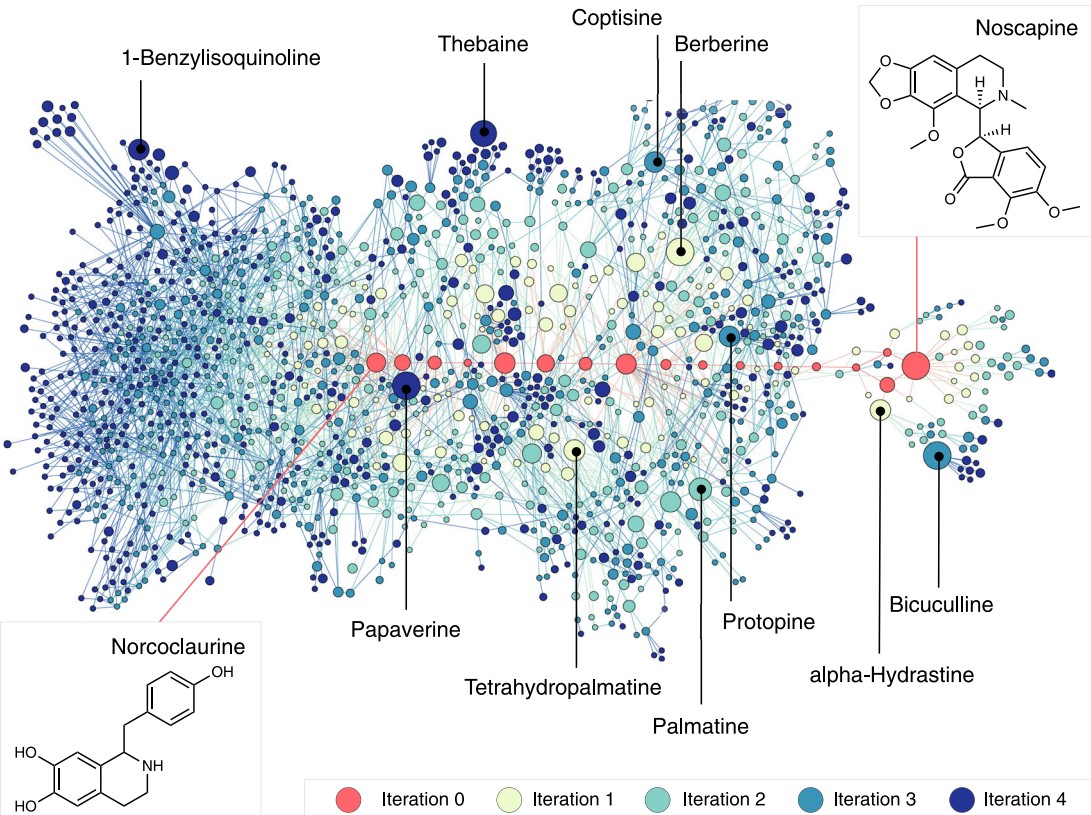

**Fig. 2 Visualization of the expanded biosynthesis network of the noscapine pathway.** The nodes and edges drawn in red show the original noscapine pathway. Around the original pathway, the predicted network of compounds (nodes) and reactions (edges) is visualized. The top 10 compounds in terms of popularity (total number of patents plus citations) are named and localized on the map. The color of the nodes shows in which iteration the compound was generated in the network reconstruction process, which is also the number of reaction steps between the original pathway and the compound. The size of the nodes is proportional to the popularity. The molecular structure of the pathway precursor, norcoclaurine, and the final product, noscapine, are shown.

**Table 1 Compounds ordered by descending popularity that are one reaction step away from intermediates in the noscapine pathway.**

| Popularity rank | Name | Best BridgIT score | Predicted EC | Number of citations | Number of patents | Citations + patents |
|---|---|---|---|---|---|---|
| 1 | Berberine | 1.00 | 1.3.3.8 | 5430 | 6751 | 12154 |
| 2 | Tetrahydropalmatine | 1.00 | 2.1.1.89 | 530 | 355 | 885 |
| 3 | Columbamine | 0.99 | 1.3.3.8 | 131 | 235 | 366 |
| 4 | Salutaridine | 1.00 | 1.14.19.67 | 85 | 264 | 349 |
| 5 | Norlaudanosoline | 0.99 | 1.14.14.102 | 144 | 177 | 321 |
| 6 | Stepholidine | 0.78 | 1.14.13.31 | 157 | 140 | 297 |
| 7 | Allocryptopine | 0.32 | 1.14.13.239 | 111 | 159 | 270 |
| 8 | Laudanine | 1.00 | 2.1.1.291 | 23 | 112 | 135 |
| 9 | Codamine | 0.79 | 2.1.1.121 | 13 | 61 | 74 |
| 10 | Norreticuline | 0.09 | 1.5.3.10 | 33 | 40 | 73 |
| 11 | Corytuberine | 0.56 | 1.14.19.67 | 18 | 39 | 57 |
| 12 | Lambertine | 0.45 | 1.3.1.29 | 30 | 23 | 53 |
| 13 | Armepavine | 1.00 | 2.1.1.291 | 28 | 15 | 43 |
| 14 | 1,2-Dehydroreticuline | 1.00 | 1.5.1.27 | 3 | 40 | 43 |
| 15 | Nandinine | 1.00 | 1.14.19.73 | 1 | 39 | 40 |

bioactive, and the remaining 1419 as chemical compounds (Supplementary Table 1). The compounds in the network were connected by 7527 reactions, of which 49 were known to be catalyzed by well-characterized enzymes linked to a genetic sequence from at least one organism in our reference database, the Kyoto Encyclopedia of Genes and Genomes (KEGG)[30]. As an additional validation, we collected 13 secondary metabolites with known biosynthesis pathways around the noscapine pathway from external sources, and we could show that all but one were found in the BIA network (Supplementary Table 2).

Our network expansion was nonuniform across the noscapine biosynthetic pathway (Fig. 2). The upstream portion of the network is highly connected, whereas the downstream portion near noscapine is less populated. This likely results from the downstream intermediates and their derivatives increasing in size and complexity, complicating their experimental detection and structural characterization. Consequently, these compounds are less represented in biological or chemical databases, and therefore are not part of the predicted network despite their increased diversity of functional groups.

**A ranking algorithm for candidate molecules highlights well-studied compounds**. To guide experimental efforts toward interesting targets for bioproduction, the numerous candidate compounds were ranked and filtered. To focus on compounds with broader interest to biomedical researchers, we ranked the candidates by popularity, defined here as the sum of the number of citations and patents reported. We screened the 1501 potential target compounds (1518 satisfying the BIA requirement minus the 17 in the noscapine pathway) and found that 204 returned at least one citation, while 467 had at least one associated patent. In total, at least one annotation (citation or patent) was obtained for 545 distinct compounds (Supplementary Data 3, Supplementary Fig. 1).

Sorting the compounds by popularity, we found that papaverine was ranked highest, with 22,918 annotations, followed by bicuculline and berberine with 16,118 and 12,154 total annotations, respectively. While the citation count reflects scientific interest in a compound, the number of patents indicates its commercial applications. As an example, the compound bicuculline, which ranked first in citations but fourth in patents, is widely employed in medical research to mimic epilepsy in mammals[43], but has a relative lack of clinical applications.

**Computational pathway construction identifies tetra-hydropalmatine as a high-priority target**. While the application

of a ranking algorithm to the potential compounds generated by BNICE.ch identifies top candidates, it does not prioritize those which can be feasibly produced experimentally. To maximize the probability of successful in vivo production of a target molecule, we applied additional filters to determine the best candidates for bioproduction. Four criteria were considered: (i) one or more production pathways toward the target compound are thermodynamically feasible; (ii) enzymes are available which natively perform similar transformations; (iii) the target compound is only one chemical transformation from an intermediate in the original pathway to focus experimental efforts on a single enzymatic step; and (iv) the target molecule is a potential or confirmed pharmaceutical.

We first examined the biological feasibility of the potential pathways to our target compounds. For the top 50 ranked candidates, we enumerated all possible pathways connecting a noscapine pathway intermediate to each target within a maximum of four reaction steps. Reaction directionalities with a highly positive standard Gibbs free energy of reaction (i.e., reactions producing molecular oxygen, binding carbon dioxide to the substrate, or demethylating the substrate via $S$-adenosylhomocysteine) were excluded to avoid thermodynamic and catalytic bottlenecks. We identified feasible pathways for 42 of 50 targets, furnishing a total of 1338 pathways (Supplementary Data 4). All of the proposed pathways are listed and visualized online (https://lcsb-databases.epfl.ch/pathways/GraphList).

To assess the availability of enzymes to catalyze the proposed reactions, we predicted enzymes for each novel reaction step using BridgIT[23]. BridgIT calculates a reactive-site centric similarity score (BridgIT score) between the novel reaction and a reference database of known, well-characterized reactions (KEGG). The output is a ranked list of candidate enzyme classes and associated similarity scores that indicate the probability that members of the candidate enzyme class will catalyze the novel reaction. As an overall metric for compound feasibility, we used the mean of the top BridgIT scores of each reaction in the pathway (available as part of the pathway visualization online).

We next examined the distance (i.e., number of reaction steps) of the target compounds from the original pathway. We restricted our search to candidates that are only one reaction from an intermediate, resulting in 15 candidates, each produced by a feasible reaction and associated with a ranked list of predicted, putative enzymes (Table 1, Supplementary Table 3). The highest ranked candidate was berberine, for which a heterologous biosynthetic pathway has already been established[44]. We therefore selected the second highest ranked candidate, (S)-tetrahydropalmatine, for

experimental validation. (S)-Tetrahydropalmatine naturally occurs in a number of plants, especially those in the genus *Corydalis* and *Stephania rotunda*, which are traditionally used in Chinese herbal medicine[45]. (S)-Tetrahydropalmatine (i.e., levo-tetrahydropalmatine) has been used for its analgesic, anxiolytic, and sedative effects as an alternative to opiates and benzodiazepines, and has shown promise in treating opiate, cocaine, and methamphetamine addiction[29].

**BridgIT analysis indicates top enzyme candidates for tetrahydropalmatine bioproduction.** Once a compound of interest is chosen, enzyme(s) catalyzing the desired transformation must be identified. BridgIT identifies known enzymes whose native reactions most closely resemble our desired reaction, and the BridgIT similarity score can be used to rank the candidates by their likelihood to catalyze the desired transformation.

(S)-Tetrahydropalmatine can be produced in one step via methylation of the 2-hydroxyl of the noscapine pathway intermediate (S)-tetrahydrocolumbamine with concomitant conversion of S-adenosylmethionine to S-adenosylhomocysteine (Fig. 3a). Because of the lack of sequence annotation for this reaction in KEGG, we used the BridgIT data described above to identify candidate enzymes. The BridgIT analysis produced a list of enzyme classes ranked by their BridgIT scores, measuring the structural similarity of the (S)-tetrahydrocolumbamine methylation to the native reactions of those enzymes (Table 2). Enzymes without protein sequence annotation were removed.

The top enzyme classes yielded promising candidates for in vivo testing. The first candidate, reticuline 7-O-methyltransferase (EC 2.1.1.291), has a BridgIT score of 0.98, making it a good candidate for in vivo testing; one variant occurs in *Papaver somniferum*. Ranked second (BridgIT score of 0.76) is the enzyme columbamine O-methyltransferase (EC 2.1.1.118; variant from *Coptis japonica* referred to here as CjColOMT), which converts (S)-columbamine to (S)-palmatine, a similar reaction to our target reaction. A literature search showed that CjColOMT has previously been found to exhibit promiscuous activity in vitro on (S)-tetrahydrocolumbamine[46]. However, while KEGG catalogues the methylation of (S)-tetrahydrocolumbamine to produce (S)-tetrahydropalmatine, it does not link it to CjColOMT or any other known gene sequence.

The analysis further showed that the O-methyltransferases (OMTs) in the noscapine pathway are among the top-ranked candidates for catalyzing the predicted reaction. It has been shown that the majority of metabolic reactions are catalyzed by promiscuous enzymes[47], and enzymes that participate in specialized metabolism are even more likely to be promiscuous[48–50]. The potential promiscuity of the noscapine biosynthetic enzymes is thus unsurprising, especially if promiscuous activity is seen on other pathway intermediates that necessarily resemble their native substrates structurally. The enzymes 6OMT (EC 2.1.1.128) and 4'OMT (EC 2.1.1.116), which O-methylate the noscapine pathway intermediates (S)-norcoclaurine and (S)-3'-hydroxy-N-methylcoclaurine, respectively, are ranked third and fourth, with BridgIT scores of 0.75. The enzyme S9OMT (EC 2.1.1.117) is ranked tenth with a BridgIT score of 0.64. The high BridgIT scores associated with these three enzymes indicate their potential for promiscuous activity on (S)-tetrahydrocolumbamine. As variants of these three enzymes are already present in the noscapine pathway prior to (S)-tetrahydrocolumbamine, their potential to produce (S)-tetrahydropalmatine will necessarily be evaluated in vivo.

**Two predicted enzymes enable tetrahydropalmatine production in vitro and in vivo.** The preceding workflow generates a ranked list of candidate enzymes predicted to produce the target product. Validation of candidate enzymes can be performed in vitro and/or in vivo in the context of a heterologous pathway. The ranking of potential enzymes enables a smaller set of enzymes to be tested experimentally, thereby maximizing the success of the project.

We selected seven of the top 18 hits from BridgIT for experimental validation, with the objective to sample a broad range of BridgIT scores. As described above, three of these enzymes—Ps6OMT, Ps4'OMT, and PsS9OMT—are already present in the biosynthetic pathway upstream of (S)-tetrahydrocolumbamine. The other four enzymes were selected based on the diversity of their native substrates, which span a range of less than 300 Da (2,4',7-trihydroxyisoflavanone) to greater than 900 Da (caffeoyl-CoA) (Table 2). These four candidate enzymes—columbamine OMT from *Coptis japonica* (CjColOMT, ranked second), O-demethylpuromycin OMT from *Streptomyces albroniger* (SaPurOMT, ranked 9th), 2,4',7-Trihydroxyisoflavanone OMT from *Lotus japonica* (LjFlaOMT, ranked 11th), and caffeoyl-coenzyme A OMT from *Arabidopsis thaliana* (AtCafOMT, ranked 17th)—were codon optimized for expression in *S. cerevisiae*, cloned into high-copy plasmids, and transformed into a de novo (S)-tetrahydrocolumbamine-producing *S. cerevisiae* strain. (S)-Tetrahydropalmatine was produced in every strain tested (Fig. 3b). However, the strain expressing the highest ranked candidate of those tested, CjColOMT, produced eight-fold more (S)-tetrahydropalmatine relative to an empty plasmid control. We hypothesized that the background (S)-tetrahydropalmatine in all strains was due to one or more of the other methyltransferases present in the heterologous (S)-tetrahydrocolumbamine-producing strain. As these enzymes' native substrates are precursors of, and structurally similar to, (S)-tetrahydrocolumbamine, they may possess promiscuous activity on (S)-tetrahydrocolumbamine itself. In fact, the other four pathway methyltransferases— S9OMT (acts natively on (S)-scoulerine), CNMT (acts natively on coclaurine), 6OMT (acts natively on norcoclaurine), and 4'OMT (acts natively on 6-methyl-(S)-laudanosoline)—were assigned high scores by BridgIT for their potential activity on (S)-tetrahydrocolumbamine, further supporting this hypothesis.

We next tested each pathway methyltransferase in vitro to determine their contribution to the background (S)-tetrahydropalmatine production. In the originally constructed heterologous (S)-tetrahydrocolumbamine pathway, the four methyltransferases were derived from *Papaver somniferum*, and thus were named Ps6OMT, PsCNMT, Ps4'OMT, and yPsS9OMT (the y prefix on the lattermost denotes that it has been codon optimized for expression in the yeast *S. cerevisiae*). Ps6OMT, PsCNMT, yPsS9OMT, and CjColOMT expressed well in *E. coli*, but no conditions tested afforded soluble Ps4'OMT. Accordingly, we examined 4'OMT variants from other species and codon optimized three for expression in *E. coli*—Cj4'OMT from *Coptis japonica*, Ec4'OMT from *Eschscholzia californica*, and Tf4'OMT from *Thalictrum flavum*. These variants expressed well in *E. coli* and were purified for in vitro analysis. As these 4'OMT variants might not possess the same substrate promiscuity as the variant originally tested (Ps4'OMT), we created strains with Ps4'OMT replaced with each alternative 4'OMT codon optimized for expression in *S. cerevisiae*. We verified that, in each of these strains, (S)-tetrahydropalmatine was still observed and that expression of CjColOMT resulted in 3- to 7-fold increased production of (S)-tetrahydropalmatine (Fig. 3c).

We tested each pathway methyltransferase and CjColOMT in vitro to determine which convert (S)-tetrahydrocolumbamine to (S)-tetrahydropalmatine. In vitro reactions were performed with Ps6OMT, PsCNMT, Cj4'OMT, Ec4'OMT, Tf4'OMT, yPsS9OMT, and CjColOMT. Ps6OMT, PsCNMT, and the 4'OMT variants produced no (S)-tetrahydropalmatine in vitro

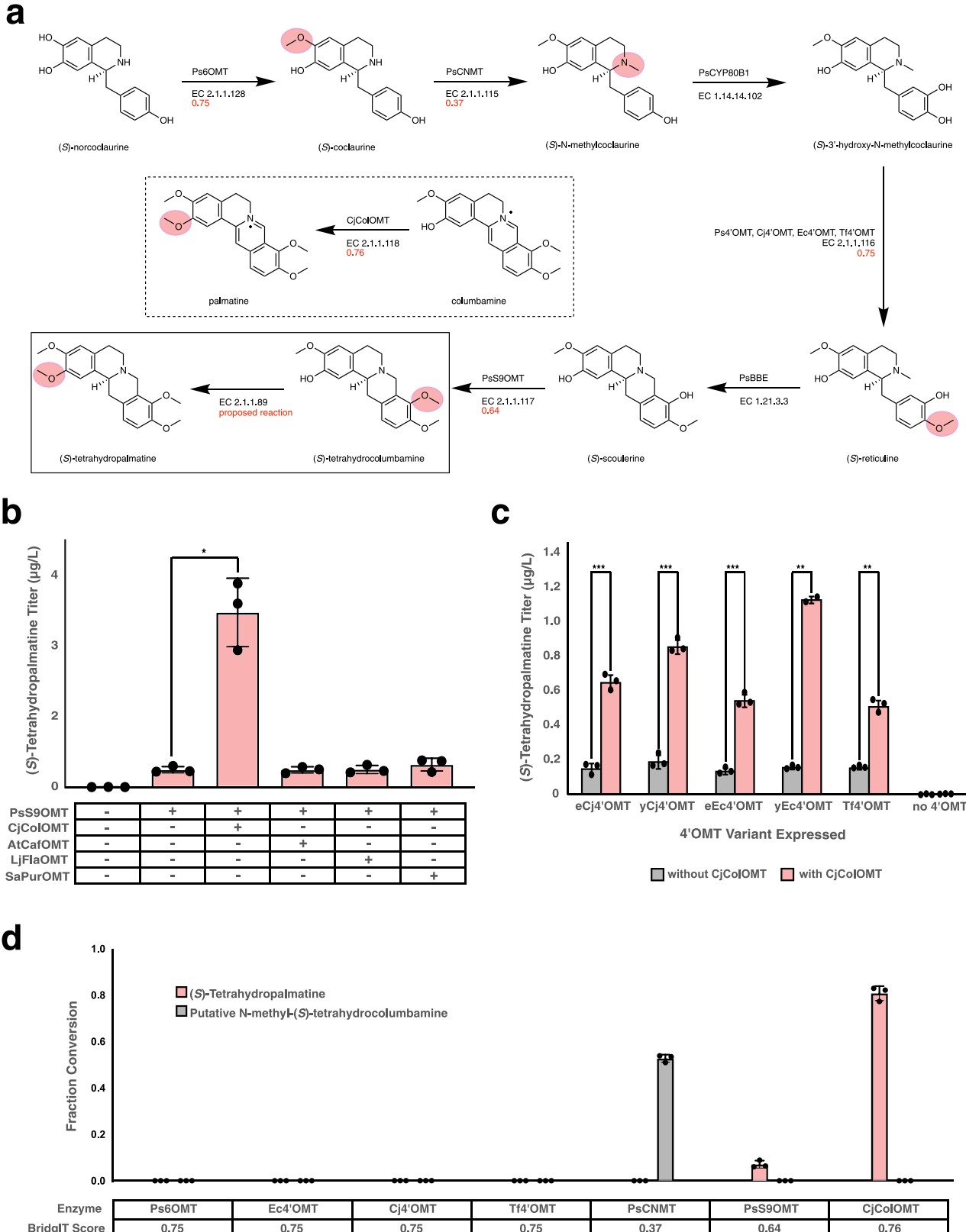

(Fig. 3d). While PsCNMT does accept (*S*)-tetrahydrocolumba-mine as a substrate, the product is presumably the *N*-methylated derivative, as the mass is consistent with a second methylation event, no (*S*)-tetrahydropalmatine production was observed, and the *N*-position is the only other available site likely to be methylated by a methyltransferase. Of the pathway enzymes, only

*Ps*S9OMT produced (*S*)-tetrahydropalmatine in vitro and thus is likely the sole source of the background (*S*)-tetrahydropalmatine observed in vivo. To further support this hypothesis, a strain lacking both y*Ps*S9OMT and CjColOMT produced no (*S*)-tetrahydropalmatine (Fig. 3b). When tested in vitro, CjColOMT afforded over 11-fold higher conversion of (*S*)-

**Fig. 3 In vivo and in vitro activity of predicted enzymes. a** Biosynthetic pathway from (S)-norcoclaurine, the first dedicated intermediate in the pathway, to (S)-tetrahydropalmatine. The specific enzyme(s) used in our strains are indicated above each reaction arrow, while below each arrow is the enzyme class and, for methyltransferases, the BridgIT score (in red) obtained for the likelihood of members of that class to perform our proposed reaction. Our proposed reaction, the methylation of (S)-tetrahydrocolumbamine to afford (S)-tetrahydropalmatine, is shown in the box at the bottom left. Shown in dotted lines is the native reaction of CjColOMT, the enzyme which was predicted and demonstrated to perform our proposed reaction. The site of methylation of each methyltransferase is highlighted on its product in pink. **b** De novo production of (S)-tetrahydropalmatine in yeast strains engineered to express members of the two most downstream O-methyltransferase classes (S9OMT & ColOMT) predicted by BNICE.ch & BridgIT to accept (S)-tetrahydrocolumbamine as a substrate. PsS9OMT is integrated into the yeast genome, while CjColOMT, AtCafOMT, LjFlaOMT, and SaPurOMT were expressed from a high-copy plasmid; the first two strains shown contain an empty version of this plasmid. Strains were cultured in selective media (YNB-Ura) with 2% dextrose, 2 mM L-DOPA, and 10 mM ascorbic acid at 30 °C for 120 h before LC-MS/MS analysis of the growth media. Data are presented as mean values ± the standard deviation of three biologically independent samples. Asterisks represent Student's two-tailed t-test: *$p < 0.05$, **$p < 0.01$, ***$p < 0.001$. Exact p-values are given in Supplementary Table 8. **c** In vitro reactions of purified methyltransferases on (S)-tetrahydrocolumbamine to produce (S)-tetrahydropalmatine (shown in pink) or the putative N-methyl-(S)-tetrahydrocolumbamine product (shown in gray). BridgIT score denotes the score obtained by BridgIT for the enzyme class to which each enzyme belongs. Data are presented as mean values ± the standard deviation of three biologically independent samples. Asterisks represent Student's two-tailed t-test: *$p < 0.05$, **$p < 0.01$, ***$p < 0.001$. Exact p-values are given in Supplementary Table 8. **d** De novo production of (S)-tetrahydropalmatine in yeast strains engineered to express alternative 4'OMTs. Strains were cultured in selective media (YNB-Ura) with 2% dextrose, 2 mM L-DOPA, and 10 mM ascorbic acid at 30 °C for 72 h before LC-MS/MS analysis. Data are presented as mean values ± the standard deviation of three biologically independent samples. Source data underlying Fig. 3**b**–**d** are provided as a Source Data file.

tetrahydrocolumbamine to (S)-tetrahydropalmatine than yPs-S9OMT (Fig. 3d), which is consistent with the significantly higher production of (S)-tetrahydropalmatine in vivo upon expression of CjColOMT (Fig. 3b). CjColOMT also accepted (S)-scoulerine as a substrate and performed methylations at both available hydroxyl groups, producing (S)-tetrahydrocolumbamine, (S)-tetrahydropalmatrubine, and (S)-tetrahydropalmatine (Supplementary Fig. 2).

**Our workflow guided production of three additional derivatives in vivo.** In order to demonstrate the generality of our approach, we applied our workflow to the prediction and experimental validation of three additional derivatives of intermediates in the noscapine pathway. We reexamined our list of the top candidate products that are one enzymatic step away from an intermediate in the noscapine pathway and that possess a high combined number of associated citations and patents (Table 1, Supplementary Table 3) and selected three products for experimental validation: (S)-armepavine, (S)-laudanine, and (S)-nandinine (Fig. 4a). (S)-Armepavine and (S)-laudanine can each be produced via regioselective O-methylations of the 7-hydroxyl groups of (S)-N-methylcoclaurine and (S)-reticuline, respectively, while (S)-nandinine can be produced from (S)-scoulerine through methylenedioxy ring formation from vicinal hydroxyl and methoxyl groups performed by a cytochrome P450. The top five candidate enzymes to perform each of these three reactions were determined by BridgIT (Supplementary Tables 4 and S5). The same five enzymes—CjColOMT, CjN6OMT, Ps7OMT, PsHNC4'OMT, and Ps6OMT (this last enzyme was already integrated into both parent strains, but was also tested on a high-copy plasmid as it was one of the top hits identified by BridgIT)—were the top candidates for both (S)-armepavine and (S)-laudanine biosynthesis, likely because both are the result of N-methylations of structurally similar substrates. For (S)-nandinine biosynthesis, the top five candidate enzymes were AmCYP719A13, EcCYP719A3, NnSCNS, CYP719A21, and ShCYP719A23. Each gene was codon optimized for expression in S. cerevisiae (Supplementary Data 5), cloned into a high-copy plasmid, and transformed into the strain that produces the substrate of its predicted product de novo: for (S)-armepavine biosynthesis (Fig. 4b), a de novo (S)-N-methylcoclaurine strain (CSY1322) was used as the parent; for (S)-laudanine biosynthesis (Fig. 4c), a de novo (S)-reticuline strain (CSY1171); and for (S)-nandinine biosynthesis (Fig. 4d), a de novo (S)-scoulerine strain (CSY1320) (Supplementary Table 6). Each candidate strain was grown and then analyzed by LC-MS/MS for production of the

predicted products; for each product, two of the enzymes tested were found to produce the desired derivative.

**Discussion**

In silico tools for novel biosynthetic pathway design can guide and accelerate metabolic engineering to produce molecules of interest. In this work, we employed the biochemical reaction prediction tool BNICE.ch[12] to explore potential biosynthesis targets that can be produced from the noscapine pathway. While multiple pathway prediction tools have been reported, most extract reaction rules automatically from biochemical databases[15–18], risking the propagation of errors (e.g., unbalanced, orphan or hypothetical multistep reactions) from database entries to the rules. In contrast, BNICE.ch rules are created manually to ensure that the predicted reactions follow biochemical logic. Furthermore, typical retrobiosynthetic approaches focus on a single predetermined compound, whereas our workflow quickly identifies a large number of candidate molecules without requiring prior knowledge of their identities. The high number of available tools stands in contrast to the small number of reported experimental validations of novel, predicted reactions. The first successfully predicted novel bioproduction pathway was established for 1,4-butanediol[51] using the commercial tool SimPheny which, like BNICE.ch, relies on expert-curated generalized reaction rules. Furthermore, novel reactions predicted by BNICE.ch in the ATLAS of Biochemistry[52,53], a repository of hypothetical biochemical reactions, have only recently been experimentally tested and validated[54]. Both of the examples of successful implementation of predicted novel reactions to date have utilized expert-curated reaction rules.

Once a pathway has been designed, enzymes need to be found to catalyze the predicted biotransformations. Available tools for enzyme function prediction determine the structural similarity of the desired reaction's reactants and products to substrates and products of known enzymes[23–26]. In contrast to other tools, BridgIT incorporates information encoded in the BNICE.ch reaction rules to identify the reactive site and then examines the atom connectivity around the reactive sites of the known and desired substrates. While all mentioned tools benchmarked their predictive capacity on datasets of known enzyme-reaction pairings, no direct experimental validation of an enzyme prediction tool has been reported to our knowledge.

In this study, BNICE.ch identified 15 potential compounds that are one reaction step from an intermediate of the noscapine biosynthetic pathway. We chose to rank these compounds by the

**Table 2 Reaction similarities between the predicted tetrahydropalmatine-producing reaction and its top 18 most similar, gene-annotated reactions from the BridgIT reference database.**

| Rank | BridgIT score | Predicted EC | Native substrate | Type of substrate | Native organism | Enzymes tested | Activity on THCB |
|---|---|---|---|---|---|---|---|
| 1 | 0.98 | 2.1.1.291 | (S)-reticuline | BIA | P. somniferum | | Not tested |
| **2** | **0.76** | **2.1.1.118** | **Columbamine** | **BIA** | **Coptis japonica** | **CjColOMT** | **Active** |
| 3 | 0.75 | 2.1.1.128 | (S)-norcoclaurine | BIA | P. somniferum | Ps6OMT | Already in pathway, no activity |
| 4 | 0.75 | 2.1.1.116 | 3'-Hydroxy-N-methyl-(S)-coclaurine | BIA | P. somniferum | Ps4'OMT | Already in pathway, no activity |
| | | | | | C. japonica<br>E. californica<br>T. flavum | Cj4'OMT<br>Ec4'OMT<br>Tf4'OMT | |
| 5 | 0.72 | 2.1.1.146 | Isoeugenol | Phenylpropanoid | Ocimum basilicum (basil) | | Not tested |
| 6 | 0.69 | 2.1.1.38 | O-demethylpuromycin | Antibiotic | Streptomyces alboniger | SaPurOMT | No activity |
| 7 | 0.68 | 2.1.1.6 | Catechol | Phenol | Diverse | | Not tested |
| 8 | 0.66 | 2.1.1.212 | 2,4',7-Trihydroxy-isoflavanone | Flavanonoid | Lotus japonica | LjFlaOMT | No activity |
| 9 | 0.64 | 2.1.1.4 | N-acetylserotonin | Neurotransmitter | Homo sapiens (human) | | Not tested |
| **10** | **0.64** | **2.1.1.117** | **(S)-Scoulerine** | **BIA** | **P. somniferum** | **yPsS9OMT** | **Already in pathway, active** |
| 11 | 0.63 | 2.1.1.231 | 4'-Hydroxyflavone | Flavonoid | Glycine max (soybean) | | Not tested |
| 12 | 0.63 | 2.1.1.68 | (E)-caffeate | Phenylpropanoid | Diverse | | Not tested |
| 13 | 0.62 | 2.1.1.104 | Caffeoyl-CoA | Phenylpropanoid | Arabidopsis thaliana | AtCafOMT | No activity |
| 14 | 0.61 | 2.1.1.150 | (E)-caffeate | Phenylpropanoid | Medicago sativa (alfalfa) | | Not tested |
| 15 | 0.61 | 2.1.1.222 | 3-Demethylubiquinol | Quinone | Diverse bacteria | | Not tested |
| 16 | 0.60 | 2.1.1.279 | trans-Anol | Phenol | Pimpinella anisum (anise) | | Not tested |
| 17 | 0.60 | 2.1.1.94 | 16-Hydroxytabersonine | Terpene indole alkaloid | Catharanthus roseus | | Not tested |
| 18 | 0.59 | 2.1.1.114 | 3,4-Dihydroxy-5-all-trans-polyprenylbenzoate | Quinone | Diverse, incl. S. cerevisiae | | Natively present in yeast |

In bold: Enzymes that act on (S)-tetrahydrocolumbamine (THCB).
EC Enzyme Commission classification number.

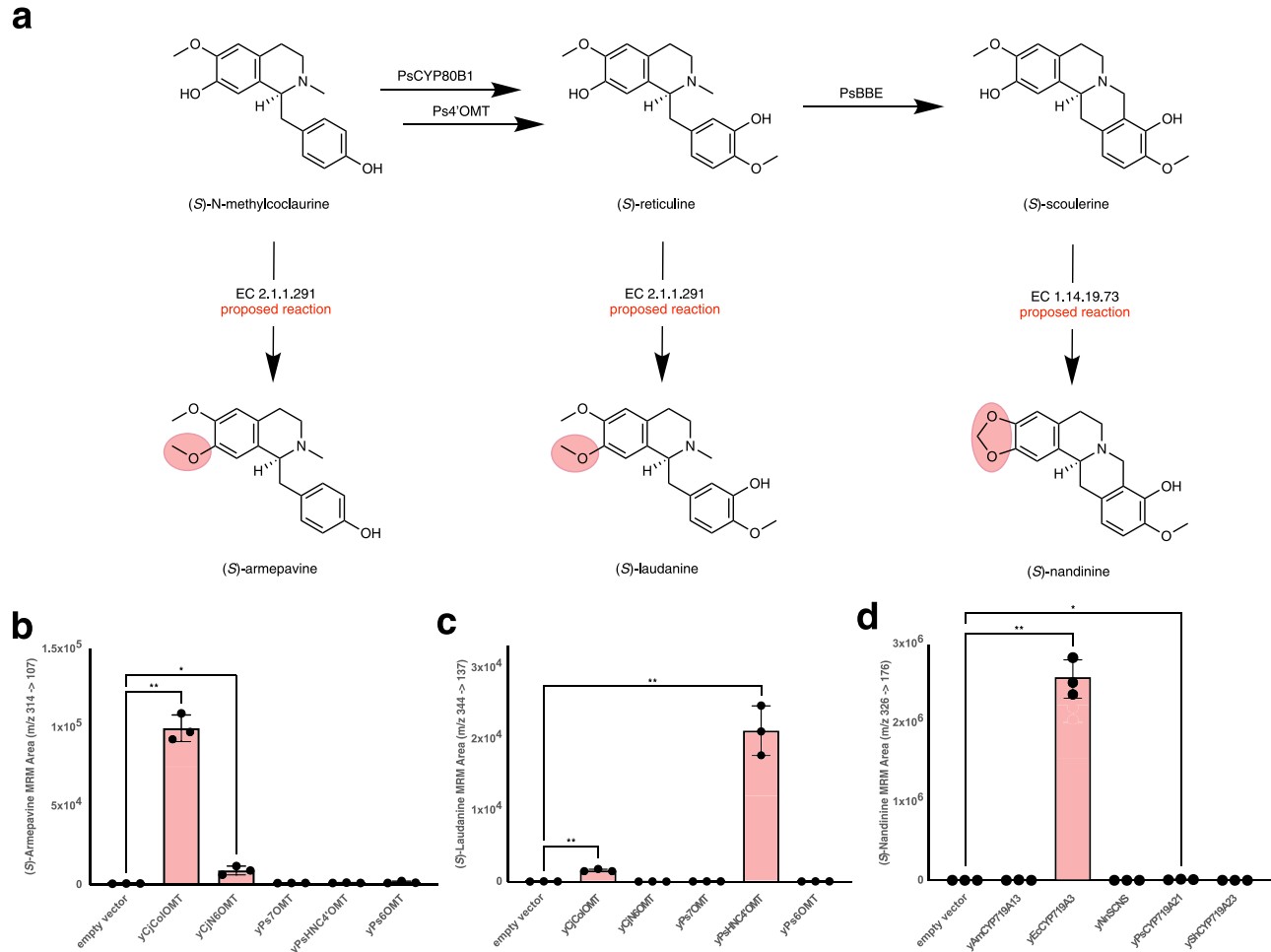

**Fig. 4 Pathway expansion to produce three additional derivatives de novo. a** Portion of noscapine biosynthetic pathway from (*S*)-*N*-methylcoclaurine to (*S*)-scoulerine with potential additional enzymatic steps to produce three derivatives: (*S*)-armepavine, (*S*)-laudanine, and (*S*)-nandinine. For steps upstream of (*S*)-*N*-methylcoclauine, downstream of (*S*)-scoulerine, or the structure of the omitted intermediate ((*S*)-3′-hydroxy-*N*-methylcoclaurine), see Fig. 3a. De novo production of the three derivatives are shown in panels **b** ((*S*)-armepavine production), **c** ((*S*)-laudanine production), and **d** ((*S*)-nandinine production). The *y*-axes of each graph in panels **b**–**d** show the integrated area of the peak measured by LC-MS/MS multiple reaction monitoring (MRM) at the quantifier transition indicated for each compound (see Supplementary Table 7 for additional details). For each derivative, all genes necessary for biosynthesis of the substrate are integrated into the genome of the parent strain (CSY1322 for (*S*)-armepavine production; CSY1171 for (*S*)-laudanine production; CSY1320 for (*S*)-nandinine production), while the genes encoding the enzymes predicted by BridgIT (shown on the *x*-axes of the graphs in panels **b**–**d**) are expressed from a high-copy plasmid. Strains were cultured in selective media (YNB-Ura) with 2% dextrose, 2 mM L-DOPA, and 10 mM ascorbic acid at 25 °C for 96 h before LC-MS/MS analysis of the growth media. Data are presented as mean values ± the standard deviation of three biologically independent samples. Asterisks represent Student's two-tailed *t*-test: *$p < 0.05$, **$p < 0.01$, ***$p < 0.001$. Exact *p*-values are given in Supplementary Table 8. Source data underlying Fig. 4**b**–**d** are provided as a Source Data file.

sum of their reports in the scientific literature and patents, in order to identify compounds of known biological interest. Alternatively, the ranking could be reversed to identify compounds for which no scientific reports are available, and to engineer the biosynthesis of these molecules to facilitate their further characterization. Literature-based ranking has the advantage that it can be used without defining a strict objective prior to the analysis, as it is the case in our study. For other applications of our workflow, alternative ranking algorithms could be used; for example, if searching for drug candidates, Lipinski's rule of five[55] could be employed, prioritizing compounds over a given molecular mass, calculated partition coefficient, and/or number of hydrogen bond donors and acceptors. In a similar way, text mining techniques could be used to retrieve associations of a given compound with clinical data or pharmaceutical studies from literature. One could also prioritize the potential compounds' chemical novelty in order to most

effectively leverage the biosynthesis platform to manufacture molecules that cannot be synthesized chemically. Additional information could be integrated into our workflow as well; for example, transcriptomics data could be used to analyze a PNP biosynthetic pathway with enzymes missing for one or more steps. BNICE.ch could determine which enzyme classes are likely to perform the missing steps, then BridgIT could rank all enzymes that are co-expressed with known pathway enzymes to determine which candidates should be investigated by virus-induced gene silencing (VIGS) or functional assays.

The top two compounds in our ranking that are one biosynthetic step from a noscapine pathway intermediate were berberine and (*S*)-tetrahydropalmatine. The heterologous biosynthesis of berberine has been previously reported[56]; however, the final reaction in its biosynthesis in this strain occurs spontaneously, as the enzyme thought to carry out its biosynthesis in plants appears to be inactive in *S. cerevisiae*, as is a related enzyme[57]. We

therefore chose to focus our efforts on (S)-tetrahydropalmatine, as numerous methyltransferases have been reported to be active in *S. cerevisiae*, thus decreasing the likelihood that we would encounter false negatives due to lack of expression or proper folding. We recently reported the de novo heterologous biosynthesis of (S)-tetrahydropalmatine in *S. cerevisiae* via an engineered variant of TfS9OMT, a homologue of PsS9OMT from *Thalictrum flavum*[58]. In particular, the biosynthesis of (S)-tetrahydropalmatine was observed with one of two native isoforms of TfS9OMT tested at a level of 0.7 μg/L, and was then increased over fivefold via structure-guided engineering, ultimately yielding a titer of 3.60 μg/L. In contrast, using BridgIT we identified a scouleline 9-O-methyltransferase (PsS9OMT) and a columbamine O-methyltransferase (CjColOMT) that both perform this transformation, and their expression together in *Saccharomyces cerevisiae* led to a titer of 3.45 μg/L using only native, non-engineered enzymes, nearly matching the titer reported with the best engineered TfS9OMT variant. Replacement or supplementation of PsS9OMT with the engineered TfS9OMT variant could increase our titer of (S)-tetrahydropalmatine, or active-site mutagenesis, as was performed for TfS9OMT, could enhance the activity of PsS9OMT or CjColOMT.

The ability of CjColOMT and PsS9OMT to methylate (S)-tetrahydrocolumbamine may seem unsurprising, as the native substrates of both enzymes are chemically similar to (S)-tetrahydrocolumbamine (Fig. 3a). In fact, both of these enzymes have been reported to have promiscuous activity toward (S)-tetrahydrocolumbamine in vitro[59]; however, these non-native activities were not available in our reference database (KEGG). While KEGG includes an entry on the conversion of (S)-tetrahydrocolumbamine to *(S)*-tetrahydropalmatine, this is an orphan reaction with no gene or protein sequence associated with it. Recent studies have indicated that 40–50% of all reactions catalogued in KEGG are orphan reactions[60,61]. In some of these cases, non-native activity data may be available, but is buried in literature and not readily accessible via existing databases, and thus might be overlooked by or unavailable to researchers. In such cases, our computational workflow can provide predictions to guide researchers to enzyme candidates to investigate further, both experimentally and in the existing literature. Furthermore, in cases where the desired non-native enzyme activities have not been reported, our workflow has demonstrated the capability to infer likely off-target activity from only native enzyme data.

The biosyntheses of three additional derivatives—(S)-armepavine, (S)-laudanine, and (S)-nandinine—demonstrated the generality of our workflow and also highlighted interesting details of some of the evaluated enzymes. CjColOMT, one of the enzymes predicted by BridgIT (and verified experimentally) to convert (S)-tetrahydrocolumabine to (S)-tetrahydropalmatine (*vide supra*) also converted (S)-*N*-methylcoclaurine to (S)-armepavine (Fig. 4b) and (S)-reticuline to (S)-laudanine (Fig. 4c), indicating that this enzyme is able to accept and functionalize a range of BIA substrates. Furthermore, CjColOMT also accepted (S)-scouleline as a substrate, methylating both open hydroxyl groups on opposite ends of the substrate (Supplementary Fig. 2). Despite the chemical similarity of (S)-*N*-methylcoclaurine and (S)-reticuline, strikingly different activities were observed with the other candidate methyltransferases tested: N6OMT from *C. japonica* (CjN6OMT) was found to accept (S)-*N*-methylcoclaurine as a substrate, producing (S)-armepavine, but demonstrated no such activity on (S)-reticuline to produce (S)-laudanine, while HNC4'OMT from *P. somniferum* (PsHNC4'OMT) showed the exact opposite activities. For (S)-nandinine biosynthesis, both CYP719A3 from *E. californica* (EcCYP719A3) and CYP719A21 from *P. somniferum* (PsCYP719A21) accepted (S)-scouleline as a substrate to produce (S)-nandinine (Fig. 4d), though with very

different activities; the activity of EcCYP719A3 is over 100-fold higher than that of PsCYP719A21. As was the case for the activities of CjColOMT and PsS9OMT on (S)-tetrahydrocolumbamine, EcCYP719A3[62] has been previously shown to accept (S)-scouleline as a substrate in vitro. Conversely, the CYP719A13 from *Argemone mexicana* (AmCYP719A13[63]) was also shown to accept (S)-scouleline as a substrate in vitro, while we saw no such activity in vivo, and PsCYP719A21[64] was reported to not accept (S)-scouleline as a substrate in vitro, while we did see activity in vivo. Taken together, these results indicate differences in activity seen for these enzymes under different reaction conditions, whether in vitro or in vivo, and the value of having a rapid in silico determination of top enzyme candidates for experimental validation.

This work serves as a proof-of-concept that our computational workflow can use a heterologous biosynthetic pathway to identify a series of potential products and the enzymes required to make those products, thus generating a starting point for subsequent optimization. Protein engineering can then be employed to substantially increase the activity of the integrated enzyme, as has been demonstrated for many classes of enzymes in the past[65–67]. Recent years have seen a dramatic increase in the complexity of biosynthetic pathways expressed in heterologous hosts[4], as well as in the efficiency with which these pathways have been reconstructed, spurred by advances in DNA synthesis, sequencing, analytical techniques, and methods for genetic engineering. As increasing numbers of heterologous biosynthetic pathways become available to the research community, as they have for such diverse compound classes as noscapinoids[5], opioids[68], flavonoids[69,70], cannabinoids[71], and carotenoids[72], computational tools to leverage these pathways for the production of additional products of interest will become increasingly useful. As the number of reported enzymes and compounds also increases, reflected by the continuous growth of biochemical databases like KEGG, we anticipate that computational tools will play a vital role in leveraging this vast amount of data to drive engineering efforts toward the bioproduction of valuable chemicals and pharmaceuticals.

## Methods

**Computational exploration of the biochemistry surrounding the noscapine pathway**. The computational workflow consists of three steps: (i) expansion of a biochemical reaction network around the original pathway, (ii) popularity assessment of compounds via annotation and ranking, and (iii) feasibility assessment via reaction annotation, pathway assembly, and pathway evaluation. The output of the computational analysis was directly used for the design of engineered yeast strains.

**Expansion of a biochemical network**. A hypothetical biochemical network using BNICE.ch[12] was expanded around the input pathway, consisting of 17 metabolites connected by 17 reactions and catalyzed by a total of 11 generalized reaction rules, using a collection of 442 bidirectional generalized enzymatic reaction rules. In a first iteration, the integrated network generation algorithm applies the reaction rules on the input molecular structures (MDL molfiles), which generates all biochemically possible reactions according to the reaction mechanisms represented in BNICE.ch. The products of these reactions are stored, and used as input compounds for the next iterations of reaction generation. This iterative process generates hypothetical biochemical networks around any given set of input molecules.

BNICE.ch distinguishes between known and novel compounds by looking up the generated molecular structures in different databases: if the compound is part of any biological, bioactive, or chemical database it is considered as known and annotated with the corresponding database identifiers. The following databases are used: the Kyoto Encyclopedia of Genes and Genomes (KEGG)[30], SEED[31], HMDB[32], MetaCyc[33], Brenda[73], MetaNetX[35], Rhea[36], BiGG[74], PMN[38], KNApSAcK[39] for biological compounds, ChEBI[40] and ChEMBL[41] for bioactive compounds, and PubChem[42] for chemical compounds. In this workflow, only known molecular structures are allowed in the network generation. Reactions are classified as known if they are part of the KEGG reaction database or the noscapine pathway, and as novel if they are not.

**Compound annotation and ranking**. We assessed the popularity of the generated compounds in the second step of the workflow by determining how many times

each compound appears in scientific publications, and how many patents are associated with the molecule. The number of publications was derived from PubChem and PubMed, while the number of patent annotations was extracted from PubChem. We used the PUG-REST service to retrieve information on compounds from the PubChem website (https://pubchem.ncbi.nlm.nih.gov/)[75] on the number of associated patents and citations. We also used the Entrez Programming Utilities (E-utilities) API service to search the PubMed database for citations by compound name[76]. We only kept compounds with at least one annotation as potential targets for biosynthesis.

**Reaction annotation and pathway ranking.** To determine if the potential targets for biosynthesis have valid bioproduction pathways, we listed all possible pathways connecting any noscapine pathway intermediate to the potential target within a maximum of four reaction steps. The pathway search algorithm NICEpath was employed to extract linear pathways from the network of reactant-product pairs[77]. The maximal number of reaction steps allowed in a pathway was set to 4, and all possible pathways connecting the noscapine pathway intermediate to the target compound were extracted. Reaction directionalities producing molecular oxygen and reverse decarboxylations were excluded from the pathway search because of their high energy demand. Also, demethylation reactions producing S-adenosylmethionine from S-adenosylhomocysteine were not allowed (other demethylation transformations are allowed).

To find enzymes for the predicted reactions in each pathway, we used the enzyme prediction tool BridgIT[23]. BridgIT calculates a similarity score between the novel reaction and reactions from a reference database of known, enzyme-annotated reactions (KEGG reaction database, downloaded in February 2018) by comparing the molecular fingerprints on and around the reactive sites of the participating reactants. The similarity between fingerprints is expressed as a score ranging from zero (no similarity) to one (the two reactions are identical up to seven atoms around the reactive site). A BridgIT score above 0.3 is considered as significant. For each reaction in the pathways, we performed BridgIT and we collected all the reactions from the reference database that had a score of 0.3 or higher. From the top score of each reaction in the pathway, we calculated the average to provide an overall metric for the enzymatic feasibility of the pathway. The pathways are available online including the top five enzymes predicted by BridgIT and associated similarity scores.

**Validation of predicted BIA network on known biosynthesis pathways.** To validate the biochemical relevance of the BNICE.ch network prediction, we collected all known secondary metabolites and their biosynthetic pathways around noscapine pathway intermediates. The test metabolites had to fulfill the following criteria: (i) The compound is 4-reaction steps away from an intermediate, or it is the endpoint of the biosynthesis pathway. (ii) The compound fits the definition of a simple BIA (single BIA unit of at least 16 carbon atoms, 13 hydrogen atoms, and 1 nitrogen atom). (iii) The compound is not a stereoisomer of a noscapine pathway intermediate, since BNICE.ch does not consider stereochemistry. We found seven compounds in the KEGG map 'Isoquinoline alkaloid biosynthesis' that matched the criteria, and another eight compounds in the MetaCyc Pathways Class 'Isoquinoline and Benzylisoquinoline Alkaloid Biosynthesis'. For each of the 13 compounds in the unified list we determined whether or not its biosynthesis pathway is present in the predicted BIA network (Supplementary Table 2).

**Yeast strain construction.** Strains used in this work are listed in Supplementary Table 6. All strains used are derived from the previously reported strain CSY1171[58]. Strains were grown nonselectively in yeast-peptone media supplemented with 2% w/v dextrose (YPD media), yeast nitrogen base (YNB) defined media (Becton, Dickinson and Company, BD) supplemented with synthetic complete amino acid mixture (YNB-SC; Clontech) and 2% w/v dextrose, or on agar plates made using the aforementioned media. Strains transformed with plasmids bearing the URA3 auxotrophic selection marker were grown selectively in YNB media supplemented with 2% w/v dextrose and uracil (YNB-Ura; Clontech) or on YNB-Ura agar plates.

Yeast genomic modifications were performed using the CRISPRm method[78]. Oligonucleotides used in this work (Supplementary Data 6) were synthesized by the Stanford Protein and Nucleic Acid Facility (Stanford, CA). Biosynthetic genes used in this study (Supplementary Table 5) were codon optimized using GeneArt Gene Optimizer software (Thermo Fisher Scientific) for expression in S. cerevisiae or E. coli (Supplementary Data 5) and then synthesized as either gBlock DNA fragments (Integrated DNA Technologies, IDT) or gene fragments (Twist Bioscience). All biosynthetic genes were synthesized with overhangs on both the 5′ end (5′—TCGACGGATTCTAGAACTAGTGGATCCTATACA—gene—3′) and 3′ end (5′—gene—TAGCCATAAGAATTCAGACACTCGAGAACTCA—3′) for ease of cloning. CRISPRm plasmids expressing Streptococcus pyogenes Cas9 (SpCas9) and a single guide RNA (sgRNA) targeting a locus of interest in the yeast genome were constructed by assembly PCR and Gibson assembly of DNA fragments encoding SpCas9 (pCS3410), tRNA promoter and HDV ribozyme (pCS3411), a 20-nt guide RNA sequence (synthesized by the Stanford Protein and Nucleic Acid Facility), and tracrRNA and terminator (pCS3414)[79]. For gene insertions, integration fragments containing the gene(s) of interest flanked by a promoter and

terminator were constructed by PCR amplification such that they possessed 40 bp overhangs on either end with homology to the yeast genome surrounding the site targeted by the guide RNA sequence. Approximately 300 ng of each integration fragment was co-transformed with 300 ng of the CRISPRm plasmid expressing the sgRNA targeting the desired genomic site. Positive integrants were identified by yeast colony PCR, DNA sequencing (Quintara Biosciences; South San Francisco, CA), and/or functional screening by LC-MS.

**Plasmid construction.** Plasmids used in this study (Supplementary Data 7) were constructed through Gibson assembly. Gibson assembly was performed by amplifying both the gene of interest and the destination plasmid (pCS952[68] or pET28) with 40 bp homologous overhangs. PCR amplifications were performed using Q5 DNA polymerase (NEB) and linear DNA fragments were purified using the DNA Clean and Concentrator-5 kit (Zymo Research). Assembled plasmids were propagated in chemically competent E. coli (TOP10; Thermo Fisher Scientific) using heat-shock transformation and selection on Luria-Bertani (LB)-agar plates with carbenicillin (100 μg/mL; for pCS952 derived plasmids) or kanamycin (50 μg/mL; for pET28 derived plasmids). Plasmid DNA was isolated by alkaline lysis from overnight E. coli cultures grown at 37 °C and 250 rpm in selective LB media using Econospin columns (Epoch Life Science) according to the manufacturer's protocol.

**Yeast transformations.** Yeast strains were chemically transformed using the Frozen-EZ Yeast Transformation II Kit (Zymo Research). Individual colonies were inoculated into YPD media and grown overnight at 30 °C and 250 rpm. Saturated cultures were back-diluted into three new cultures at 1:5, 1:10, and 1:20 dilutions in YPD media and grown for an additional 5–7 h to reach exponential phase. For each transformation, 1 mL aliquots from each back-diluted culture were pelleted by centrifugation at 500 × g for 4 min (successively pelleting aliquots from each different dilution into a single pellet in a 1.5 mL microcentrifuge tube) and then washed twice by resuspending the pellet in 1 mL of 50 mM Tris-HCl buffer, pH 8.5. Washed pellets were resuspended in 50 μL of EZ2 solution per transformation and mixed with 100–600 ng of total DNA and 500 μL of the EZ3 solution. The yeast suspensions were incubated at 30 °C with gentle inversion for one hour. For plasmid transformations, the transformed yeast was directly plated onto YNB-Ura agar plates. For Cas9-mediated gene integrations, the yeast suspensions in the EZ3 solution were first mixed with 1 mL YPD media, pelleted by centrifugation at 500 × g for 4 min, and then resuspended in 250 μL of fresh YPD media. The suspensions were incubated at 30 °C with gentle inversion for an additional 90 min to allow production of G418 resistance proteins and then spread onto YPD plates containing 400 mg/L G418 sulfate. For all transformations, plates were incubated at 30 °C for 72 h before being used to inoculate cultures for metabolite assays.

**Growth conditions for metabolite assays.** Metabolite production tests were performed in YNB-SC or YNB-Ura media with at least three replicates. Yeast colonies were inoculated into 300 μL of media and grown in 2 mL deep-well 96-well plates covered with AeraSeal gas-permeable film (Excel Scientific). Cultures were then grown for 72–120 h at 25 °C or 30 °C (exact duration and temperature are specified in each figure), 460 rpm, and 80% relative humidity in a Lab-Therm LX-T shaker (Adolf Kuhner).

**Analysis of metabolite production.** Cultures were pelleted by centrifugation at 3500 × g for 5 min at 4 °C and 100 μL aliquots of the supernatant were removed for direct analysis. Metabolite production was analyzed by LC-MS/MS using an Agilent 1260 Infinity Binary HPLC and an Agilent 6420 Triple Quadrupole mass spectrometer. Chromatography was performed using a Zorbax EclipsePlus C18 column (2.1 × 50 mm, 1.8 μm; Agilent Technologies) with water with 0.1% v/v formic acid as solvent A and acetonitrile with 0.1% v/v formic acid as solvent B. The column was operated with a constant flow rate of 0.4 mL/min at 40 °C and a sample injection volume of 5 μL. Compound separation for the detection of (S)-tetrahydropalmatine, (S)-laudanine, or (S)-nandinine was performed using the following gradient: 0.00–0.10 min, 10% B; 0.10–5.00 min, 10–40% B; 5.00–5.50 min, 40% B; 5.50–6.00 min, 40–98% B; 6.00–10.00 min, 98% B; 10.00–10.01 min, 98–10% B; 10.01–13.00 min, equilibration with 10% B. Compound separation for the detection of (S)-armepavine was performed using the following gradient: 0.00–0.20 min, 10% B; 0.20–10.00 min, 10–14% B; 10.00–11.00 min, 14–90% B; 11.00–12.00 min, 90% B; 12.00–12.20 min, 90-10% B; 12.20–12.70 min, equilibration with 10% B. The LC eluent was directed to the MS from 1–10 min operating with electrospray ionization (ESI) in positive mode, source gas temperature 350 °C, gas flow rate 11 L/min and nebulizer pressure 40 psi. Metabolites were quantified by integrated peak area in MassHunter Workstation software (Agilent) based on the multiple reaction monitoring (MRM) parameters in Supplementary Table 7. Integrated peak areas were converted to titers by comparison to standard curves prepared using a commercial standard of (S)-tetrahydropalmatine (Toronto Research Chemicals). Primary MRM transitions for (S)-tetrahydropalmatine were identified by analysis of a 0.1 mM standard in methanol using the MassHunter Optimizer software package (Agilent); all other MRM transitions used were previously reported and are described in Supplementary Table 7.

**Enzyme expression and purification.** Plasmids containing the gene of interest in a pET28 expression vector (see Supplementary Data 7 for a full list of plasmids used in this study) were used to transform *E. coli* BL21(DE3) (Invitrogen) competent cells containing the pGro7 chaperone expression plasmid (Takara) via heat shock. Briefly, 1 ng of plasmid DNA was added to a 50 μL aliquot of competent cells, the tube was chilled on ice for 15 minutes, placed in a 42 °C water bath for 35 seconds, then returned to ice for 2 min. Seven hundred fifty μL of SOC media were then added and the tube was rotated at 37 °C for 45 min before being plated on an LB agar plate containing 50 μg/mL kanamycin and 20 μg/mL chloramphenicol. A single colony was then picked and used to inoculate a primary culture of 5 mL of LB media containing 50 μg/mL kanamycin and 20 μg/mL chloramphenicol which was then grown for 24 h. Five hundred microliter of this primary culture were then used to inoculate a secondary or expression culture of 50 mL of TB medium containing 50 μg/mL kanamycin and 20 μg/mL chloramphenicol (for all proteins except PsS9OMT) or 500 mL of LB medium containing 50 μg/mL kanamycin (for PsS9OMT). This expression culture was grown to an $OD_{600}$ of 0.6–1.0 and then induced with IPTG (for *O*-methyltransferase induction, GoldBio) and L-arabinose (for groES/groEL induction, Fischer Scientific) at final concentrations of 0.1 mM and 2 mg/mL, respectively, for all proteins except PsS9OMT, which was induced with only IPTG to a final concentration of 1 mM. The expression culture was then grown at 30 °C (for all proteins except PsS9OMT) or 16 °C (for PsS9OMT) for 20 h at 250 rpm, after which the culture was harvested by centrifugation (10 minutes at $3500 \times g$ in a 50 mL Falcon tube) and stored at −20 °C until lysis and purification.

Frozen pellets were then thawed and resuspended in 25 mL of Ni-nitrilotriacetic (Ni-NTA) equilibration buffer (50 mM sodium phosphate, 300 mM NaCl, 10 mM imidazole, pH 7.4) and lysed by sonication while kept on ice (Branson Sonifier 450, 0.5" horn, 50% duty cycle, $4 \times 1$ min with 2 min rests). Lysed cultures were then clarified by centrifugation (45 min at $35,000 \times g$ at 4 °C) and the clarified lysate was purified by Ni-NTA affinity chromatography. Briefly, 1 mL of Ni-NTA resin (Fisher Scientific) was equilibrated with at least 5 volumes of Ni-NTA equilibration buffer (described above) and then loaded with the clarified lysate. The loaded resin was then washed with at least 5 volumes of Ni-NTA wash buffer (50 mM sodium phosphate, 300 mM NaCl, 50 mM imidazole, pH 7.4) and then the bound protein was eluted with 5 volumes of Ni-NTA elution buffer (50 mM sodium phosphate, 300 mM NaCl, 250 mM imidazole, pH 7.4). The eluted fractions were then combined and concentrated using an Amicon® 30 kDa cutoff spin filter (EMD Millipore) at $5000 \times g$ at 4 °C. Concentrated protein fractions were then exchanged into storage buffer (50 mM potassium phosphate, 100 mM NaCl, 10% glycerol, pH 7.5), split into separate aliquots, and stored at −20 °C until use.

**In vitro bioconversions.** Analytical reactions were carried out at the 50 μL scale in triplicate. To a 1.5 mL Eppendorf tube were added 5 nmol substrate (final concentration of 100 μM; (*S*)-norcoclaurine and (*S*)-scoulerine purchased from Toronto Research Chemicals; norlaudanosoline purchased from Santa Cruz Biotechnology), 1 μmol sodium ascorbate (final concentration of 25 mM), 5 nmol *S*-adenosylmethionine (SAM, final concentration of 100 μM; purchased from Sigma-Aldrich), and 150 pmol purified methyltransferase enzyme (3 μM final concentration) in 50 mM potassium phosphate, pH 8.0. The reactions were shaken at 600 rpm at 37 °C for 2 h before being quenched with an equal volume of methanol, spun down at $20,000 \times g$ for 10 min, and filtered prior to LC-MS analysis (see 'Analysis of metabolite production' section above for details on LC-MS analysis conditions).

**Metabolite purification.** The large scale in vitro (*S*)-scoulerine conversion reaction was carried out on the 20 mg scale at a final reaction volume of 610 mL in a 2 L Erlenmeyer flask. To this flask were added 20 mg (61 μM) of (*S*)-scoulerine (final concentration of 100 μM), 15 mmol sodium ascorbate (final concentration of 25 mM), 61 μM SAM (final concentration of 100 μM), and 73 nmol of purified TfS9OMT DS M111A[58] (final concentration of 0.12 μM) in 50 mM potassium phosphate, pH 8.0. The reaction was incubated at 37 °C at 250 rpm. The reaction was ultimately run for 15 h, but was monitored to ensure conversion had stopped by analytical LC-MS. To do so, 50 μL aliquots were pulled periodically, quenched with an equal volume of MeOH, spun down at $20,000 \times g$ for 10 min, and filtered prior to LC-MS analysis (see 'Analysis of metabolite production' section above for details on LC-MS analysis conditions).

Once the reaction was complete, 30 g of Amberlite XAD4 resin were added and the flask was shaken overnight at 30 °C at 250 rpm. The Amberlite XAD4 resin was transferred to 50 mL Falcon tubes, the supernatant was decanted off, and 50 mL total MeOH were added to the four tubes containing resin. The resin in MeOH was then vortexed for 10 minutes, after which it had turned yellow. The MeOH was then pipetted into a 500 mL round-bottomed flask and was concentrated by rotary evaporation to ~2 mL, which was then pipetted into 4 tared 1.5 mL Eppendorf tubes and concentrated to dryness overnight on a speedvac. Approximately 400 mg of crude material were obtained from this process, which were then resuspended in $H_2O$ to a final concentration of 100 mg crude material/mL. This material was then purified by preparative LC (Agilent 1200 Series LC) with a Varian Pursuit XRs C18 $250 \times 10$ mm column, 5 μm particle size (solvent A = $H_2O$ with 0.1% FA, solvent B = ACN with 0.1% FA). The following LC method was used: 0–4.0 min, 20% B, 2.0 mL/minute; 4.0–12.0 min, 20–100% B, 2.0 mL/min; 12–20 min, 100% B, 2.0 mL/min;

4 minute postrun. Fractions were analyzed by LC-MS to determine, which contained the desired products (see 'Analysis of metabolite production' section above for details on LC-MS analysis conditions). Fractions containing the desired product were concentrated and re-purified by preparative LC until the desired purity was obtained.

**Software.** MarvinView was used for displaying chemical structures. Marvin 6.2.2, ChemAxon (http://www.chemaxon.com). The graph visualization tool Gephi was used to visualize metabolic networks[80].

**Reporting summary.** Further information on research design is available in the Nature Research Reporting Summary linked to this article.

## Data availability

Data supporting the findings of this work are available within the paper and its Supplementary Information files. A reporting summary for this Article is available as a Supplementary Information file. All unique biological materials generated in this study are available from the authors upon reasonable request. Additional source data beyond those presented in this paper and the Supplementary Information files are available from the corresponding authors upon request. The pathway visualization platform is available at https://lcsb-databases.epfl.ch/pathways/GraphList upon request. Source data are provided with this paper.

## Code availability

A publicly accessible version of the BNICE.ch framework including user instructions can be found at https://lcsb-databases.epfl.ch/Atlas2. The platform is freely available to academia upon request. BridgIT is available as an online tool at https://lcsb-databases.epfl.ch/Bridgit.

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

## Acknowledgements

We thank A. Cravens for the yeast CRISPRm Cas9/sgRNA plasmids (pCS3410, 3411, 3414, and 3702), T. Valentic for the pET28/Tf4'OMT plasmid (pCS4661) and the de novo *N*-methylcoclaurine strain (CSY1322), and O. Jamil for the primers for the amplification of the pCS952 vector backbone. We thank P. Srinivasan and B. Kotopka for valuable feedback in the preparation of the manuscript. Funding for this work was provided by National Institute of Health Grants (grant to C.D.S., AT007886, fellowship to J.T.P., F32 AT009509-03), the Firmenich Stanford-EPFL Exchange Program, the Swiss National Science Foundation (SNSF) (Grant number 2013/158, MicroScapesX) and the Ecole Polytechnique Fédérale de Lausanne (EPFL). The contents of this publication are solely the responsibility of the authors and do not necessarily represent the official views of NIGMS or NIH.

## Author contributions

J.P., J.H., V.H., and C.D.S. conceived of the project and wrote the manuscript. J.H. and V.H. designed the computational workflow. J.H. implemented the workflow, performed the reaction network prediction and analyzed the results. H.M. designed and performed the enzyme prediction. J.P. and C.D.S. designed the experiments and analyzed the experimental results. J.P. performed the in vivo and in vitro experiments.

## Competing interests

The authors declare no competing interests.
