## [Peer Review File · Nature Communications]

REVIEWER COMMENTS

Reviewer #1 (Remarks to the Author):

The manuscript entitled "A computational workflow for the expansion of heterologous biosynthetic pathways to natural product derivatives" presents a computational framework to identify enzymatic reactions able to catalyze the synthesis of specific compounds. In particular, this work focus in molecules derived from the noscapine pathway.

Strong points: The workflow is of high interest for the metabolic engineering community. It is nice to see that the algorithm proposed the most commercially relevant metabolites close to a pathway of interest (I am wondering if the date of the publications/patents could be taken into account, as some metabolites that were expensive or interested in the past may now be cheap or useless today). This can inform novel bioprocesses when using strains that have been well optimized for large scale production of compounds but also help to identify new pathways and enzymatic activities. The authors determine an arbitrary set of rules to narrow down the vast amount of potential enzymes and compounds, which seem to work well in the given example.

Weaker points: The majority of the workflow is based in previously existing tools such as BNICE.ch and BridgIT where these two methods are connected by a set of rules defined by the authors to simplify their experimental validation. My major comment is that the methodology is described as a general approach that can be applied to any PNP and the title and discussion is around that. However, the methodology could be apply to any other metabolite of any origin, beyond PNP. In any case, it is only applied in a pathway, a metabolite and a reaction, and validated for a particular metabolite and a particular reaction. I think that in order to keep the scope of the article as wide as it is, the workflow should be applied to other pathways, metabolites and reactions. It does not need to be something very complicated but otherwise, it is arguable that the universality of the approach is not proven, which can be especially criticized since the found enzymatic activity was already known. The case of berberine is interesting, as it is not only the top finding but also the last reaction step has not been yet found enzymatically and perhaps the results of this workflow could shed light on it and find the first enzyme that works in yeast. The authors argue that it is produced spontaneously, which is the same case as in the tetrahydropalmatine. I guess alternatively, the manuscript could be re-written for a narrower application case, the one studied and validated, but in that case it could be better suited for a more specialized journal.

Reviewer #2 (Remarks to the Author):

The manuscript of Hafner, Payne and coworkers describes a computational workflow guiding identification of enzymes allowing derivatizing intermediates of biosynthetic pathways of compounds of interest in order to produce new valuable molecules. While many medicines still derive from plant natural products, the supply of several drugs is severely affected by shortages notably due to overexploitation of plant resources and climate changes. In such a context, the production of these compounds through microbial cell factories heterologously expressing the corresponding biosynthetic pathways constitutes a reliable solution. This also offers the possibility of expanding the range of the produced compounds by introducing distinct enzymes catalyzing new or already known reactions. However, identification of enzymes catalyzing the expected missing decoration is still complex, time-consuming and never success-guaranteed. The group of C. Smolke already published outstanding results concerning the metabolic engineering of benzylisoquinoline alkaloid (BIA) synthesis in yeast while the group of V. Hatzimanikatis developed and published remarkable computational tools for enzyme prediction for instance. By joining their efforts, they developed in this work a computational workflow combining BNICE and BridgIT to screen the biochemical vicinity of biosynthetic pathway to derivatize intermediates and produce "selected" compounds by exploiting substrate promiscuity of enzymes. They illustrated this workflow by derivatizing intermediates from the BIA pathway to

produce tetrahydropalmitine through identification/functional characterization of predicted OMTs. On the overall, this manuscript provides interesting and useful data concerning the engineering of natural product biosynthesis. It also constitutes a powerful tool to guide elucidation of biosynthetic pathway and to synthesize compounds of interest using already identified enzymes. Experiments have been well conducted and results are conclusive. The manuscript is well written and can be understood by a wide audience.

Questions and Concerns:

- To validate their workflow, the authors chose target compound that is only one chemical transformation from an intermediate of the pathway. While it is totally sounding regarding the subsequent experimental efforts for testing, what about target compounds distant from 2 or 3 reactions? Are the predictions efficient for each step? Can the authors comment on this and illustrate with predictions for a more distant compound (no functional validation required)? In this case, can it be exemplified with another pathway? Such a result would reinforce the strength of their workflow.
- Authors excluded reactions with a high standard Gibbs free energy (reactions producing molecular oxygen...) to avoid thermodynamic bottlenecks. However, it includes many potential reactions from PNP biosynthesis. Is this statement only true for BIA pathway or is it a limit of the workflow?
- As such, this workflow can also be valorized as a complementary tool to elucidate PNP biosynthetic pathways. In this case, do functional orthologs of OMT from *Corydalis* and *Stephania rotunda* can be identified by homology searches if transcriptomic data are available ?
- Since a high substrate promiscuity exists for OMTs, does CjColOMT also methylate (S)-scoulerine to produce (S)-tetrahydrocolumbamine ?
- I did not see any reference to Figure 1 in the version of the manuscript I reviewed.

Reviewer #3 (Remarks to the Author):

The authors propose an computational approach to identify potential PNP biosynthesis pathways. The approach is based on the prediction of potential pathways which is then narrowed down by a literature search. Finally, it is evaluated on the biosynthetic pathway of noscapine.

Overall, the paper introduces an interesting approach. However, it lacks a more in-depth discussion and evaluation of the computational approach.

My main concern is in the second step of the workflow. The authors basically propose a literature search to rank the predicted compounds. An issue with this step in my opinion is that we enforce the bias in the literature. If this approach was widely used in the future, it would pick up any bias that was introduced in publications and rank compounds high that are anyway over-represented. Then by publishing results based on this approach, you would strengthen the bias, which then will influence future experiments that in return again strengthen the bias and so on. This limits the potential of identifying completely new pathways that have not been explored so far. It is a general problem when using published literature for predicting something new and discussed in literature. I would at least recommend to discuss this in the paper, at the moment I don't see it mentioned anywhere.

Another issue with a naive literature search is that you do not distinguish between positive or negative mentions of an entity. So if there is a compound mentioned in a large number of articles in the context of "compound X is a bad example for this and should not be considered", it will be highly ranked, even if literature would recommend otherwise. This might be beyond the scope of this paper, but there are (text mining) approaches in other areas to overcome this, a lot of research has been done in bioinformatics in this direction.

In line 147 the authors talk about the connectivity of the network and that the upstream part is higher connected than the downstream part. They speculate that this is due to the increasing size of the downstream intermediates. I wonder if this is not a property of a network in general and the decreasing connectivity propagates downstream. Coming from the "root", the network might break down into parts, each of these parts into more parts. So if at each step of the network, it is broken into n parts, after two steps we have n^2 , then n^3 , and so on. So it might not only be based on the chemical properties. This could be checked on the predicted pathways or synthetic pathways. There is probably more to it, having a lower probability to connect if two nodes are on further "levels" and a higher probability if they are closer to each other. In general, I suspect there is more behind it than just the complexity of the compounds. (This might be a bit nitpicking, but I find this interesting to think about.)

Another problem is a bit the seemingly ad-hoc choice of parameters that are not explained. As an example, in line 256, the authors say they selected 7 of the top 18 hits. Looking at it from a more computer science perspective, this lacks the explanation why this parameter is chosen. What happens if 10 are chosen? How do we choose it? What is the motivation? Discussing this will strongly help in reproducing the experiments with other compounds.

In line 339 the authors speculate that the ranking could also be done using other algorithms for other scenarios. But how about this exact application, would other rankings work similarly? What is the advantage of the literature-based ranking? Which others would potentially work?

A more general question is about the evaluation of the approach. The authors chose to carry out a case study, but couldn't you evaluate the approach additionally in a data-driven way? You could remove some known pathways from the data set and see if you would be able to reproduce them. It would be a bit tricky to avoid information leakage given that there might be publications out there that talk about it, but that could be taken into account. This would be interesting as it could be done in an automated manner and evaluate the approach much more general than using just a specific (nevertheless interesting) example.

I wondered what the general distribution of the number of hits in PubMed, PubChem and so on for this data set. I don't think it is given somewhere in the paper or supplementary, it would be interesting to see.

Finally, in line 437 the authors state they use "a path search algorithm" is used. Given just this information, I am not able to reproduce the results as I do not know the details of the algorithm. Please explain which algorithm is used.

Minor comment: A few references seem to be broken, see for example line 101 or line 147 (among others).

Authors response to the editor and reviewers' comments

We thank the editor and the reviewers for their valuable feedback and their constructive comments. As requested by several reviewers and by the editor, we performed additional experiments to show the universality of our approach and we modified the manuscript accordingly. Our point-by-point responses to the questions, comments and suggestions provided by the reviewers are provided in italics below.

REVIEWER COMMENTS

Reviewer #1 (Remarks to the Author):

Weaker points: The majority of the workflow is based in previously existing tools such as BNICE.ch and BridgIT where these two methods are connected by a set of rules defined by the authors to simplify their experimental validation. My major comment is that the methodology is described as a general approach that can be applied to any PNP and the title and discussion is around that. However, the methodology could be apply to any other metabolite of any origin, beyond PNP. In any case, it is only applied in a pathway, a metabolite and a reaction, and validated for a particular metabolite and a particular reaction. I think that in order to keep the scope of the article as wide as it is, the workflow should be applied to other pathways, metabolites and reactions. It does not need to be something very complicated but otherwise, it is arguable that the universality of the approach is not proven, which can be especially criticized since the found enzymatic activity was already known.

We thank the reviewer for their helpful comments. We agree with the reviewer that the workflow can be applied to molecules of any origin and is not necessarily restricted to PNPs. The universality of BNICE.ch and BridgIT has been shown in previous publications, such as the ATLAS of Biochemistry, where we have shown previously that these tools can be applied to any type of metabolic pathways. To demonstrate that our workflow is generally applicable to more molecules than just the (S)-tetrahydropalmatine example originally given, we applied this workflow to produce three additional derivatives from the noscapine pathway, thus showing that it can produce successful predictions for multiple products produced by multiple classes of enzymes. The computational results are shown in Supplementary Table S1.8, while the experimental results are described in a newly added section in the Results (“Our Workflow Guided Production of Three Additional Derivatives in vivo”), in a newly added figure (Figure 4), and an additional paragraph in the Discussion section.

Reviewer #2 (Remarks to the Author):

- To validate their workflow, the authors chose target compound that is only one chemical transformation from an intermediate of the pathway. While it is totally sounding regarding the subsequent experimental efforts for testing, what about target compounds distant from 2 or 3 reactions? Are the predictions efficient for each step? Can the authors comment on this and illustrate with predictions for a more distant compound (no functional validation required)? In this case, can it be exemplified with another pathway? Such a result would reinforce the strength of their workflow.

We thank the reviewer for this comment. The generated reaction network around the noscapine pathway is predicted by the network prediction tool BNICE.ch. Since no known pathways

(except for the noscapine biosynthesis pathway) were part of the input, we can evaluate the predictive power of the approach by evaluating how many known biosynthesis pathways have been reconstructed in the expanded, predictive reaction network.

For this evaluation, we collected known biosynthesis pathways towards BIAs around the noscapine pathway from the reference databases KEGG and MetaCyc, and we found that 12 out of 13 identified BIAs with known biosynthesis pathways (1-4 steps long) were present in the BNICE.ch-generated network. The manuscript has been modified accordingly to accommodate this additional analysis, and the corresponding results have been added to the submission as Supplementary Table S1.7.

- Authors excluded reactions with a high standard Gibbs free energy (reactions producing molecular oxygen...) to avoid thermodynamic bottlenecks. However, it includes many potential reactions from PNP biosynthesis. Is this statement only true for BIA pathway or is it a limit of the workflow?

We thank the reviewer for this comment. Instead of excluding reactions, we only removed the thermodynamically infeasible directionality of the reaction with a highly positive standard Gibbs free energy ($\Delta G_r'^{\circ} > +10$ kcal/mol). These reactions can be considered irreversible because the metabolite concentrations that would be necessary to reverse the reaction directionality are beyond the physiological constraints of the cell. We considered this step in the workflow to be an advantage rather than a limitation, as it removed infeasible reaction directionalities from the pathway reconstruction. Should future researchers wish to explore reactions that we excluded in this way, our workflow can be modified to remove part or all of this filter. We modified the following sentence in the results section to clarify this point:

“Reaction directionalities with a highly positive standard Gibbs free energy of reaction (i.e., reactions producing molecular oxygen, binding carbon dioxide to the substrate, or demethylating the substrate via *S*-adenosylhomocysteine) were excluded to avoid thermodynamic and catalytic bottlenecks.”

- As such, this workflow can also be valorized as a complementary tool to elucidate PNP biosynthetic pathways. In this case, do functional orthologs of OMT from *Corydalis* and *Stephania rotunda* can be identified by homology searches if transcriptomics data are available?

We thank the reviewer for this suggestion. While we considered the integration of transcriptomics data to be outside the scope of our demonstration of our workflow, transcriptomics data could be integrated into future studies to aid in the search for unknown enzymes in a biosynthetic pathway. We have added text discussing this to our Discussion section.

- Since a high substrate promiscuity exists for OMTs, does CjColOMT also methylate (S)-scoulerine to produce (S)-tetrahydrocolumbamine?

We thank the reviewer for this comment. At the reviewer's suggestion, we performed this reaction in vitro and determined that CjColOMT is able to accept (S)-scoulerine to produce (S)-tetrahydrocolumbamine, along with two other products. We have added this new data to our manuscript in Figure S2.2 as well as a discussion of these data in the main manuscript.

- I did not see any reference to Figure 1 in the version of the manuscript I reviewed.

We thank the reviewer for this comment. The missing reference to Figure 1 has been added to the introduction section in the beginning of the 5th paragraph.

Reviewer #3 (Remarks to the Author):

The authors propose a computational approach to identify potential PNP biosynthesis pathways. The approach is based on the prediction of potential pathways which is then narrowed down by a literature search. Finally, it is evaluated on the biosynthetic pathway of noscapine.

Overall, the paper introduces an interesting approach. However, it lacks a more in-depth discussion and evaluation of the computational approach.

My main concern is in the second step of the workflow. The authors basically propose a literature search to rank the predicted compounds. An issue with this step in my opinion is that we enforce the bias in the literature. If this approach was widely used in the future, it would pick up any bias that was introduced in publications and rank compounds high that are anyway over-represented. Then by publishing results based on this approach, you would strengthen the bias, which then will influence future experiments that in return again strengthen the bias and so on. This limits the potential of identifying completely new pathways that have not been explored so far. It is a general problem when using published literature for predicting something new and discussed in literature. I would at least recommend to discuss this in the paper, at the moment I don't see it mentioned anywhere.

We thank the reviewer for the constructive feedback. Indeed, the second step of our workflow is intended to help scientists distinguish between well-known compounds and novel, less studied molecules. For our validation, we wanted to produce a compound with a known scientific or industrial impact. However, the same approach can be used to do exactly the opposite, i.e., identifying compounds for which no literature is available, and use the biosynthesis platform for in vitro or in vivo production of the molecule, thus facilitating its further characterization (e.g., pharmaceutical effects). For this, one can simply reverse the ranking and prioritize compounds with a low popularity score. To clarify these points, we have added a short paragraph to the discussion section.

Another issue with a naive literature search is that you do not distinguish between positive or negative mentions of an entity. So if there is a compound mentioned in a large number of articles in the context of "compound X is a bad example for this and should not be considered", it will be highly ranked, even if literature would recommend otherwise. This might be beyond the scope of this paper, but there are (text mining) approaches in other areas to overcome this, a lot of research has been done in bioinformatics in this direction.

We agree with the reviewer that the evaluation of the efficiency and significance of text-mining approaches would be beyond the scope of the paper, but we have added the following sentence to the discussion:

"In a similar way, text mining techniques could be used to retrieve associations of a given compound with clinical data or pharmaceutical studies from literature."

Nevertheless, we would argue that even if a paper mentions the inactivity of a certain compound, it still means that this compound has been tested and has thus received scientific attention.

In line 147 the authors talk about the connectivity of the network and that the upstream part is higher connected than the downstream part. They speculate that this is due to the increasing size of the downstream intermediates. I wonder if this is not a property of a network in general and the decreasing connectivity propagates downstream. Coming from the "root", the network might break down into parts, each of these parts into more parts. So if at each step of the network, it is broken into n parts, after two steps we have n^2 , then n^3 , and so on. So it might not only be based on the chemical properties. This could be checked on the predicted pathways or synthetic pathways. There is probably more to it, having a lower probability to connect if two nodes are on further "levels" and a higher probability if they are closer to each other. In general, I suspect there are more behind it than just the complexity of the compounds. (This might be a bit nitpicking, but I find this interesting to think about.)

We thank the reviewer for the intriguing comment, and we agree that the matter is interesting to think about. We observed that the number of possible (known) derivatives decreases when we move down the pathway from norcoclaurine to noscapine. We then hypothesized that if only a few derivatives are found, this is not due to the number of biochemically possible derivatives, but to the fact that not many derivatives have been described in literature and therefore made the way to public databases (here, PubChem). To show that this is the case, we applied again the reaction rules to each intermediate in the noscapine pathway for one generation, but this time we also allowed novel compound structures to be generated by BNICE.ch. We then collected the number of known (PubChem) compounds, and compared to the number of potential derivatives generated by BNICE.ch and not known to any database. The following table shows that while the number of possible known derivatives varies significantly between the noscapine pathway intermediates, the number of hypothetical derivatives remains comparatively stable. Please note that the numbers indicate the total number of derivatives, and no specific filtering for BIAs has been applied, as it is the case in the network presented in the paper.

However, we believe that these considerations are out of the scope of our study, and thus have not included a discussion of them in the revised manuscript.

Compound identifiers		Compound derivatives (1 reaction step away)		Calculated compound properties			
KEGG ID	Compound name	Number of possible known derivatives (PubChem)	Number of hypothetical derivatives	Number of carbon atoms	Number of Hydrogen atoms	Number of Oxygen atoms	Gibbs Free Energy of formation (GCM estimations)
C06160	(S)-Norcoclaurine	21	217	16	17	3	-34.58
C06161	(S)-Coclaurine	14	439	17	19	3	-19.93
C05176	(S)-N-Methylcoclaurine	16	429	18	21	3	-7.66
C05202	3'-Hydroxy-N-methyl-(S)-coclaurine	14	451	18	21	4	-47.14
C02105	(S)-Reticuline	15	435	19	23	4	-32.89
C02106	(S)-Scoulerine	11	426	19	21	4	-34.38
C04118	Isocorypalmine	11	413	20	23	4	-19.73
C03329	(S)-Canadine	15	392	20	21	4	-21.76
C02915	(S)-cis-N-Methylcanadine	11	394	21	24	4	NaN
C21586	(S)-1-Hydroxy-N-methylcanadine	6	407	21	24	5	NaN
C21587	(13S,14R)-1,13-Dihydroxy-N-methylcanadine	7	420	21	24	6	NaN
C21588	(13S,14R)-1-Hydroxy-13-O-acetyl-N-methylcanadine	12	439	23	26	7	NaN
C21590	4'-O-Desmethyl-3-O-acetyl-papaveroxine	10	451	23	25	8	-164.77
C21600	Narcotoline hemiacetal	5	427	21	23	7	-127.73
C09593	Narcotoline	6	407	21	21	7	-127.02
C20297	Narcotine hemiacetal	5	413	22	25	7	-113.08
C09592	alpha-Narcotine	12	397	22	23	7	-112.37

Another problem is a bit the seemingly ad-hoc choice of parameters that are not explained. As an example, in line 256, the authors say they selected 7 of the top 18 hits. Looking at it

from a more computer science perspective, this lacks the explanation why this parameter is chosen. What happens if 10 are chosen? How do we choose it? What is the motivation? Discussing this will strongly help in reproducing the experiments with other compounds.

We thank the reviewer for this comment. The reason for our choice was that we wanted to sample different levels of BridgIT scores to get an overall idea of the sensitivity of the BridgIT score. However, since the objective was to produce a specific compound, it would be more appropriate from an engineer's point of view to choose the top BridgIT hits to increase the chances of successful production of the target compound. We have modified the text in the manuscript as follows to address this comment:

"We selected seven of the top 18 hits from BridgIT for experimental validation, with the objective to sample a broad range of BridgIT scores." (red: added)

In line 339 the authors speculate that the ranking could also be done using other algorithms for other scenarios. But how about this exact application, would other rankings work similarly? What is the advantage of the literature-based ranking? Which others would potentially work?

We thank the reviewer for this comment. Since the objective of our case study was to produce a compound of pharmaceutical interest, a literature-based ranking had the advantage that well-studied molecules with known pharmaceutical impact could be prioritized. Alternatively, we could have evaluated the chemical similarity of each generated compound with one or several known drug molecules, and pick those with that were structurally most similar the known drug(s). However, in this case, prior knowledge on a drug molecule of interest would be necessary, and the objective of the study would have been slightly different (i.e., find biosynthesis targets that are structurally similar to a given drug molecule). Hence, literature-based ranking has the advantage that it can be used without defining a strict objective prior to the analysis. The screening for popularity of compounds can further help the scientist to organize the data, and to learn more about the characteristics of the produced compound. In our case for example, the ranking quickly revealed which compounds are worth further investigation on their pharmaceutical activity. The discussion section has been modified to account for these considerations.

A more general question is about the evaluation of the approach. The authors chose to carry out a case study, but couldn't you evaluate the approach additionally in a data-driven way? You could remove some known pathways from the data set and see if you would be able to reproduce them. It would be a bit tricky to avoid information leakage given that there might be publications out there that talk about it, but that could be taken into account. This would be interesting as it could be done in an automated manner and evaluate the approach much more general than using just a specific (nevertheless interesting) example.

We thank the reviewer for this comment. We agree with the reviewer that a data-driven approach can help evaluate our workflow. Even though data on known biosynthesis pathways are rather sparse, we decided to evaluate the presented workflow on known biosynthesis pathways collected from KEGG and MetaCyc. Since no known pathways (except for the noscapine biosynthesis pathway) were part of the input, we could evaluate the predictive power of the approach by evaluating how many known biosynthesis pathways have been reconstructed in the expanded reaction network predicted by BNICE.ch. For this evaluation, we collected known biosynthesis pathways towards BIAs around the noscapine pathway from the reference databases KEGG and MetaCyc, and we found that 12 out of 13 identified BIAs

with known biosynthesis pathways were present in the BNICE.ch-generated network. The manuscript has been modified accordingly to accommodate this additional analysis, and the corresponding results have been added to the submission as Supplementary Table S1.7.

I wondered what the general distribution of the number of hits in PubMed, PubChem and so on for this data set. I don't think it is given somewhere in the paper or supplementary, it would be interesting to see.

We thank the reviewer for this comment. The full list of compounds and their number of patent and citation hits can be found in Supplementary Table S1.4. Additionally, we added Supplementary Figure S2.1 to visualize the distribution of annotations.

Finally, in line 437 the authors state they use "a path search algorithm" is used. Given just this information, I am not able to reproduce the results as I do not know the details of the algorithm. Please explain which algorithm is used.

We thank the reviewer for this comment. To extract pathways from the generated reaction network, we used the pathway search tool NICEpath. NICEpath uses the number of conserved atoms between a reactant and a product to create an atom-weighted graph of the reaction network, which is then searched using Yen's k-shortest loop-less path search. The name of the pathway search tool (NICEpath) and the corresponding reference to the paper have been added to the methods section (Reaction annotation and pathway ranking), as well as the parameter settings used to perform the search.

Minor comment: A few references seem to be broken, see for example line 101 or line 147 (among others).

We thank the reviewer for this comment. The broken references have been amended in the revised manuscript.

REVIEWERS' COMMENTS

Reviewer #1 (Remarks to the Author):

The authors have addressed my main comment and partially demonstrated the universality of their workflow by producing different derivatives of the (S)-tetrahydropalmatine. I would have liked to see other molecules from other pathways, which would have added more value and stronger proof.

Reviewer #2 (Remarks to the Author):

In the revised version of the manuscript, Hafner and coworkers addressed almost all the concerns raised by my previous reviewing. They notably demonstrated the efficiency of their computational workflow through the identification of other OMTs methylating new BIAs of interest and through characterization of P450s catalyzing synthesis of nandinine. One can now expect that this approach can be used to decipher distinct biosynthetic pathways of interest.

Reviewer #3 (Remarks to the Author):

First of all, I want to thank the authors for the extensive work they put into revising the manuscript and addressing my comments!

They addressed my comments in great detail and I am satisfied with the paper in this form.

Authors response to the editor and reviewers' comments

We thank the editor and the reviewers for their time and their helpful comments.

REVIEWER COMMENTS

Reviewer #1 (Remarks to the Author):

The authors have addressed my main comment and partially demonstrated the universality of their workflow by producing different derivatives of the (S)-tetrahydropalmatine. I would have liked to see other molecules from other pathways, which would have added more value and stronger proof.

We thank the reviewer for their comment. While the demonstration of our approach to other pathways is certainly of interest, we believe that this would be beyond the scope of this paper. Future work will address this broader application of this method to diverse pathways.

Reviewer #2 (Remarks to the Author):

In the revised version of the manuscript, Hafner and coworkers addressed almost all the concerns raised by my previous reviewing. They notably demonstrated the efficiency of their computational workflow through the identification of other OMTs methylating new BIAs of interest and through characterization of P450s catalyzing synthesis of nandinine. One can now expect that this approach can be used to decipher distinct biosynthetic pathways of interest.

We thank the reviewer for their comment.

Reviewer #3 (Remarks to the Author):

First of all, I want to thank the authors for the extensive work they put into revising the manuscript and addressing my comments!

They addressed my comments in great detail and I am satisfied with the paper in this form.

We thank the reviewer for their comment.